# Long-Term Exposure to Benzo[a]Pyrene Affects Sexual Differentiation and Embryos Toxicity in Three Generations of Marine Medaka (Oryzias Melastigma)

**DOI:** 10.3390/ijerph17030970

**Published:** 2020-02-04

**Authors:** Dong Sun, Qi Chen, Bo Zhu, Yu Lan, Shunshan Duan

**Affiliations:** 1Research Center of Hydrobiology, Department of Ecology, Key Laboratory of Aquatic Eutrophication and Control of Harmful Algal Blooms of Guangdong Higher Education Institute, Jinan University, Guangzhou 510632, China; jnu_sundong@163.com (D.S.); cq92088@outlook.com (Q.C.); lanyu0706@163.com (Y.L.); 2School of Life Science and Engineering, State Defense Key Laboratory of the Nuclear Waste and Environmental Security, Southwest University of Science and Technology, Mianyang 621010, China; zzhubo@126.com

**Keywords:** benzo[a]pyrene, marine medaka, multigeneration, sexual differentiation

## Abstract

Benzo[a]pyrene (BaP) is a common environmental disrupting chemical that can cause endocrine disorders in organisms. However, the continued interference effects of BaP on multi-generation fish needs further research. In this study, we performed different periods (G1F1-3, G2F2-3, G3F3) of BaP exposure on marine medaka. We determined the embryo toxicity, and analyzed relative reproductive genes (ERα, cyp19a and vtg1) to predict the sexual differentiation of marine medaka. The results showed that high concentrations of BaP (200 μg·L^−1^) significantly delayed the hatching time of embryos. Moreover, medium/high concentrations of BaP (20 and 200 μg·L^−1^) prolonged the sexual maturity time of marine medaka. The relative gene expression of ERα, cyp19a and vtg1 were measured at 5 d_pf_ of embryos. We found that BaP had significantly inhibited the expression of the genes related to female fish development. Consequently, there were more males in the offspring sex ratio at BaP exposure. Overall, BaP can cause embryonic toxicity and abnormal sexual differentiation, while the expression of related reproductive genes can effectively indicate the sex ratio.

## 1. Introduction

Benzo[a]pyrene (BaP) is one of the polycyclic aromatic hydrocarbons (PAHs) that can be found in oil, coal, fuel and many organic materials with incomplete combustion [1]. Exposure to PAHs may cause organ deformities, sex changes, decreased reproductive capacity, abnormal reproductive behavior, disorders in physiological and metabolic processes, and abnormal gene expression [2,3,4,5]. Since BaP can be biohazardous, it is classified as a human carcinogen by the International Agency for Research on Cancer (IARC) [6]. However, BaP is still widely distributed and can be detected in rivers [7], sea [8], air [9] and various organisms [10]. Its interference effects on organisms are of great concern.

BaP can influence aquatic organisms as an endocrine-disrupting chemical (EDC) [11]. Even if the concentration of BaP in the aquatic environment is very low, fish are inevitably exposed to BaP throughout their life cycle, which may disturb the normal reproductive conditions [12]. Corrales et al. [13] have found that BaP can affect the growth and development of zebrafish offspring. Meanwhile, Booc et al. [14] also found that the decreased embryos fertilization and hatching success of *Fundulus heteroclotus* were caused by BaP exposure. Moreover, some studies suggest that the expression of key genes has changed in some important pathways, which promote the toxic effects of BaP in fish, especially at critical development and reproductive stages [15]. Sex ratio is an effective biomarker used for the assessment of EDCs on reproductive interference in fish and other organisms [16]. Previous studies have reported that BaP exposure can increase the male offspring of *Tigriopus japonicus sensu lato* [17]. Contrary, Wu et al. [18] reported the increase of female individuals in marine polychaete exposed to BaP. In addition, Chikae et al. [19] reported no significant changes in the female-to-male ratio in Japanese medaka (Oryzias latipes) exposed to BaP. Therefore, the sex ratio as an environmental marker for assessment of EDCs needs to be further studied. In fact, data on the intergenerational interference and persistence of aquatic organisms exposed to BaP over prolonged periods are lacking, as well as the expression of sexual differentiation relative genes.

In the present study, BaP was used to investigate the embryonic toxicity and its underlying mechanisms in marine medaka. The BaP concentration used in our study was 2–200 μg·L^−1^, where 2 and 20 μg·L^−1^ were consistent with the environmentally relevant concentrations [20]. Hatching time, sexual maturity time and sex ratio were examined to determine the effects of the BaP on three generations of marine medaka. In addition, relative reproductive gene expression was determined in order to verify and indicate the sexual differentiation following exposure to BaP during the embryo stage.

## 2. Materials and Methods

### 2.1. Chemicals

BaP (CAS 50-32-8, purity 96%) was purchased from Aladdin (Shanghai, China). It was dissolved in dimethyl sulfoxide to obtain a 1 g·L^−1^ stock solution and stored at room temperature in the dark. The exposure concentrations were control, control+solvent (S.Control), 2, 20, and 200 μg·L^−1^, respectively. The S.Control and exposure treatment included 0.002% (*v*/*v*) dimethyl sulfoxide. 

### 2.2. Test Fish Husbandry and Reproduction

Marine medaka (*Oryzias melastigma*) was obtained from the Institute of Urban Environment of the Chinese Academy of Sciences. It was acclimated for at least one month in flow-through holding glass tanks supplied with aerated artificial seawater and 30% salinity. It was kept at 26 ± 1 °C with a photoperiod of 14 h: 10 h light: dark cycle. The fish were fed three times a day with brine shrimp (*Artemia naupii*).

Embryos hatching: marine medaka can spawn at a temperature of 25 °C. Females start spawning at 8:00–9:00 a.m. each morning (after 1 h of light onset) and can spawn 10–40 embryos. In the present study, we collected the embryos, which were incubated at 28 °C after washing away feces and bait with filtered artificial seawater. The incubation time was 9–11 days. The new seawater was changed daily and dead embryos were picked out during this period.

Juvenile fish husbandry: the juvenile fish can be fed with brine shrimp larva at 2–5 d_ph_ (days post-hatching). In order to ensure the synchronization of juvenile fish development, the stereoscopic microscope (Olympus, Tokyo, Japan) was used to select the uniformly developed embryos for exposure experiments. Other culture conditions were the same as in adult fish.

### 2.3. Exposure Assays

The F1 generation embryos with 1 d_pf_ (days post-fertilization) were exposed to different concentrations of BaP for 142 days. Fifty embryos were randomly selected in 5 L glass tanks containing 2 L of aerated seawater with BaP, performed in triplicate. After hatching, thirty juvenile fish were selected and transferred to 15 L glass tanks containing 10 L aerated seawater with BaP, performed in triplicate. Exposure period culture conditions were the same as normal feeding conditions.

The offspring of the F1 generation was continually exposed to BaP (G2F2) and non-BaP (G1F2), respectively. Similarly, the offspring of G1F2 was also exposed to non-BaP (G1F3), and the offspring of G2F2 was exposed to BaP (G3F3) and non-BaP (G2F3), respectively (Figure 1). This design was used to observe the persistence of the tendency of reproductive system interference in marine medaka at different exposure patterns of BaP. The F2 and F3 generation embryos were maintained for 145 and 158 days according to parental group under the same culture conditions as the F1 generations. Twenty embryos were homogenized and stored at −80 °C for subsequent RNA extraction in 5 dpf at each group, respectively.

The medaka was exposed to BaP at nominal concentrations of control, S.control, 2, 20 and 200 μg·L^−1^ with three replicates. Half of the treatment water was renewed daily in each tank. Dead fish in all the treatment groups were removed and recorded on a daily basis.

The sex ratio was determined based on the phenotype, and gonad morphology after anesthetization and dissection at F1–3 generations fish that reached sexual maturity.

### 2.4. Quantitative Real-Time PCR Analysis

Twenty embryos (5 d_pf_) per replicate were collected, homogenized and stored at –80 °C for subsequent RNA extraction, respectively. The Quantitative real-time PCR analysis was performed according to our previous study [21]. In brief, total RNA was extracted from three fish using Trizol reagent (Invitrogen). The total RNA quality and concentration were determined by electrophoresis on an agarose gel stained with GoldView (Beijing, China), and 260 nm by Q5000 UV-Vis spectrophotometer (Quawell, USA), respectively. RNA samples with a purity between 1.81 and 2.05 for a ratio 260/280 were used. 1 μg of total RNA was reverse-transcribed using the cDNA synthesis kit (GoScript Reverse Transcription System, Promega, Madison, USA) in a total volume of 40 μL according to the manufacturer’s instructions. The cDNA was stored at −20 °C. The qRT-PCR analysis was performed on the CFX96 Real-time system (C1000 Touch, Bio-Rad) using the GoTaq ^®^ qPCR Mater mix (Promega, USA). The qPCR reactions were initially denatured at 95 °C for 10 min, and then 40 cycles at 95 °C for 15 sec and 60 °C for 1 min. Melting curve analysis from 60 °C to 95 °C was performed to ensure the specificity of each amplicon. All primers were synthesized by BGI (Guangzhou, China; Table 1). The relative transcriptional expression levels of each gene to s.control were analyzed using 18s as a housekeeping gene to evaluate the 2^−△△Ct^ [22].

### 2.5. Data and Statistical Analysis

All figures were made using Origin 9 (OriginLab Corporation, Northampton, MA, USA), and the data analyses were performed using SPSS 19.0 software (SPSS, Chicago, IL, USA). All data were verified for normality and homogeneity of variance using the Kolmogorov–Smirnov test and Levene’s test, respectively. Group differences were compared by one-way ANOVA, followed by a Post-hoc LSD test. *p* < 0.05 was considered to be statistically significant. Data are presented as mean ± standard deviation (SD).

## 3. Results

### 3.1. Hatching Time in the Offspring of Multigeneration Marine Medaka Exposed to BaP

Hatching time was determined in the marine medaka offspring in each group exposed to BaP (Table 2). The hatching time was significantly delayed in marine medaka embryos at 200 μg·L^−1^ in each group, and also at 20 μg·L^−1^ of G1F1, G1F2 and G2F2. It is worth noting that in the G2F3 and G3F3 groups, embryos died out during hatching at 200 μg·L^−1^, which may have been due to the long-term exposure to BaP, and a large amount of BaP accumulated in the embryo that eventually led to the death of the embryo.

### 3.2. Sexual Maturity Time in Multigeneration Marine Medaka Exposed to BaP

Sexual maturity time was determined in the marine medaka from each group exposed to BaP (Table 3). There was continuous exposure to BaP in F1–3 generations, and the sexual maturation time significantly increased at 200 μg·L^−1^ in each BaP exposure group, with 2 and 20 μg·L^−1^ in the F1 generation, and 20 μg·L^−1^ in the F3 generation. However, there was also a significant increase in sexual maturation time at 200 μg·L^−1^ in the G2F3 group.

### 3.3. Sex Ratio in Multigeneration Marine Medaka Exposed to BaP

Figure 2 shows the sex ratio of marine medaka in different groups exposed to BaP. Under the BaP exposure, there was a significant increase in male marine medaka at 20 and 200 μg·L^−1^ in the F1 generation. However, there was also a significant increase in female marine medaka in the G1F3 generation of non-BaP exposure. Both G2F3 and G3F3 were significantly increased in males after BaP exposure.

### 3.4. Gene Transcriptional Expression in Multigeneration Marine Medaka Exposed to BaP

Figure 3A–C shows that after exposure at 5 dpf, ERα expression levels were significantly lower at 200 μg·L^−1^ in each group, 20 μg·L^−1^ at G2F3 and G3F3, and 2 μg·L^−1^ in each group for the F3 generation. However, there was no difference between the 20 μg·L^−1^ and S.Control in the two consecutive generations of G2F2. The cyp19a gene of 200 μg·L^−1^ was significantly lower in each group (Figure 3D–F). At the 20 μg·L^−1^, G2F2 was also significantly lower. The vtg1 gene was significantly lower compared to S.control in each BaP concentration after exposure of three consecutive generations (Figure 1), and was significantly higher in G1F3 at 200 μg·L^−1^ compared to S.control.

## 4. Discussion

BaP is a typical environmental disrupting chemical, which can potentially disrupt the normal hatching time of embryos, as well as sexual maturity and sex ratio of adult marine medaka. In this study, we found the hatching time was significantly delayed at 200 μg·L^−1^ in each group. These results were consistent with Chikae et al. [17] who reported that hatching time of harpacticoid copepod was significantly delayed at 0.01, 0.1 and 10 μg·L^-1^ of BaP exposure. We also found a significant delay at 20 μg·L^−1^ after two consecutive generations of exposure (G2F2), but no effect in G2F3, which may have been because the marine medaka reached normality through self-regulation under long-term exposure to BaP stress. It is worth noting that the offspring of G2F3 and G3F3 died during hatching after three generations of BaP exposure. These results showed that BaP might be enriched in the parental fish after continuous exposure [13,24], resulting in an increased content of BaP in the offspring or embryos, and eventually reaching a lethal level. In addition, we found significantly delayed hatching time in the F1–2 following three consecutive generations of BaP exposure group (G1F1-3), except for F3. It is possible that marine medaka becomes resistant to BaP stress after long-term exposure [21,25].

The observed high concentration of BaP (200 μg·L^−1^), can lead to prolonged sexual maturity time of marine medaka. Wu et al. [18] found that BaP can significantly inhibit the sexual maturation ratio of marine polychaete *Perinereis nuntia*. The delayed sexual maturity may be due to oxidative stress, which affects the energy stores exposure to endocrine disruptors [26]. Previous studies reported that fish biotransformation of PAHs might produce reactive oxygen species, thus leading to oxidative stress [18,27].

In the present study, we found an increase in male offspring after BaP exposure. This result showed that BaP is a typical anti-estrogen phenomenon that can increase the expression of androgen receptors in the testes and liver [28]. This is consistent with the previous study by Bang et al. [17] who reported that the male offspring were observed in harpacticoid copepod at BaP exposure. This result may also be related to the exposure concentration of BaP, considering that a previous study reported that sex ratio did not change following exposure to a low concentration of BaP [19]. Nevertheless, we found an increased female offspring in G1F3, which may be related to feedback overcorrection regulation.

EDCs can interfere with the normal endocrine system and reproductive hormones in the organism [29], and then cause changes in ERα, cyp19a and vtg genes expression levels and sex ratio occurrence [30,31]. The cyp19a and ERα genes are related to aromatase, which is involved in female organism development [32]. The vtg is the egg-yolk protein vitellin precursor that is synthesized by mature females [18]. The transcriptional expressions of cyp19a genes were downregulated after BaP exposure. This result was consistent with Hoffmann and Oris who found that BaP could alter the expression of aromatase by downregulating the cyp19 gene expression in zebrafish [33]. The ERα gene was significantly downregulated after BaP exposure. This may be because BaP increases the androgen receptor, thus causing competition for the same target between the estrogen receptor and the androgen receptor [34]. The vtg1 gene was consistent with ERα [12], which was downregulated after BaP exposure. The changes in these gene expressions can indirectly reflect and predict the population sex dynamics.

## 5. Conclusions

Even though the toxicity of BaP has been studied over many years, research on the toxicity of BaP on multigeneration and its persistence are still lacking. In this study, we found that BaP had a typical toxic effect, which could delay the hatching time of embryos, and prolong the sexual maturity time in adult marine medaka. Meanwhile, BaP also has anti-estrogen and androgenic effects, and can significantly inhibit the expression of ERα, cyp19a, and vtg1genes, which are related to female marine medaka reproductive development, and a higher production of male offspring produced from the parental marine medaka. Future studies should address persistent multigenerational deformities, lethal effects, and reproductive regulation of relative genes caused by BaP exposure.

## Figures and Tables

**Figure 1 ijerph-17-00970-f001:**
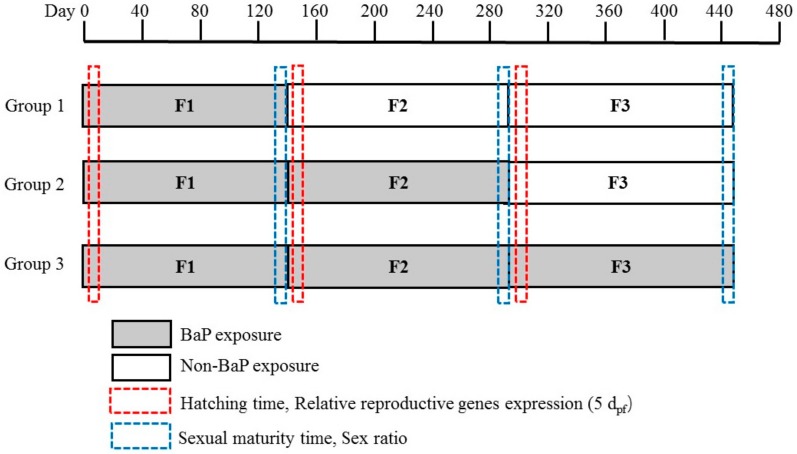
Schematic representation of experiment design.

**Figure 2 ijerph-17-00970-f002:**
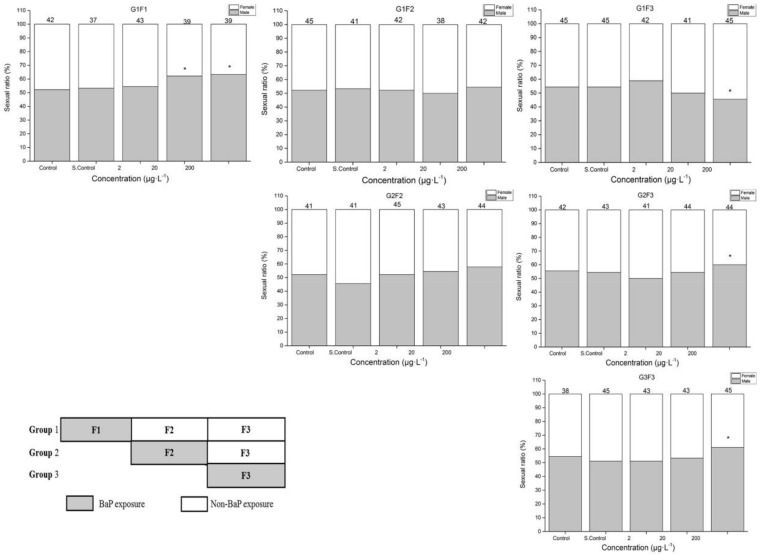
Data are expressed as the percentage of males and females in each group. A significant difference in sex ratio from control: * *p* < 0.05.

**Figure 3 ijerph-17-00970-f003:**
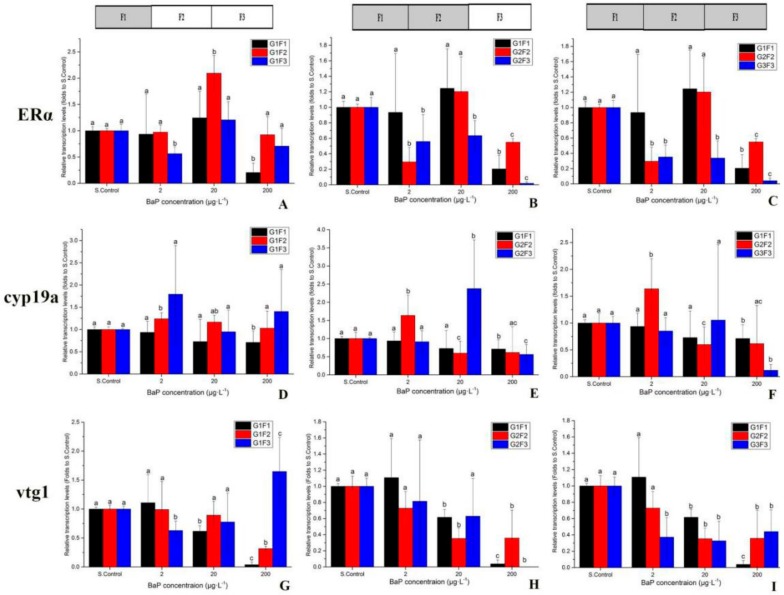
The mRNA levels of the genes in embryos after exposure to BaP for 5 d_pf_. Bars with different letters indicate statistical significance (*p* < 0.05) as assessed using one-way ANOVA followed by the LSD test. (**A**–**C**) ERα gene expression at different exposure patterns; (**D**–**F**) cyp19a gene expression at different exposure patterns; (**G**–**I**) vtg1 gene expression at different exposure patterns.

**Table 1 ijerph-17-00970-t001:** Primers for quantitative real-time PCR in marine medaka.

Gene	Sense Primer (5′-3′)	Antisense Primer (5′-3′)	GenBank Number	Reference
18s	AACGCTGTGCTGCGTAGCCTCAATT	AGAAGAAGCCCCACTTTTCCTCGCA	DQ105650	[23]
ERα	TCGCCGCTGTTGTGCTGTGATGTT	TCCTGGATCTGAGTGCGGGTCCGA	JF907629	[23]
Cyp19a	ACCTCGCGTTTTGGCAGCAAACA	TTTCCACAGCGCCACGTTGTTGT	JF907625	[23]
Vtg1	TTGGCAGAGATGCAGCAGCGGT	GGAAATGCAGGACACCCCAGTAGCC	JF268651	[23]

**Table 2 ijerph-17-00970-t002:** Hatching time of offspring in each generation in different groups of Benzo[a]pyrene (BaP) exposure.

BaP Concentration (μg·L^−1^)			Hatching Time (d_pf_)			
	G1F1	G1F2	G1F3	G2F2	G2F3	G3F3
Control	10.5 ± 0.5	11.33 ± 0.58	11.1 ± 0.17	10.33 ± 0.58	11.43 ± 0.51	11.16 ± 0.29
S.Control	10.67 ± 0.58	10.67 ± 0.58	11 ± 0.3	11.33 ± 0.58	10.33 ± 0.85	11.1 ± 1.01
2	10.33 ± 0.58	10.68 ± 0.58	9.9 ± 0.17	10.33 ± 0.58	10 ± 0.7	10.57 ± 1.25
20	11.16 ± 1.04 *	12.1 ± 0.85 *	10.67 ± 1.15	12.67 ± 0.58 *	11.67 ± 0.58	11.43 ± 0.51
200	13.67 ± 0.58 *	12.9 ± 0.85 *	13.43 ± 1.25 *	15.1 ± 1.15 *	0 *^a^	0 *^a^

*, ≤0.05, compared with the control in the same groups. ^a^, death during hatching.

**Table 3 ijerph-17-00970-t003:** Sexual maturity time of marine medaka in different groups exposed to BaP.

BaP Concentration (μg·L^−1^)			Sexual Maturity Time (dph)			
	G1F1	G1F2	G1F3	G2F2	G2F3	G3F3
Control	125 ± 1	127.33 ± 3.06	124.67 ± 14.47	121.67 ± 10.97	133 ± 2	121.33 ± 11.59
S.Control	127 ± 1.73	127.33 ± 7.09	119.33 ± 4.16	123.33 ± 6.66	131.33 ± 3.21	120.33 ± 2.08
2	131.67 ± 1.53 *	126 ± 3.61	128.67 ± 5.03	127 ± 3.61	136 ± 2	119 ± 5
20	134 ± 2 *	124 ± 5.57	122 ± 6.24	126.33 ± 6.66	132.33 ± 2.08	142.67 ± 6.11 *
200	142 ± 2.65 *	130 ± 6.08	124 ± 8	145 ± 6.56 *	149.33 ± 6.66 *	158.67 ± 8.02 *

*, ≤0.05, compared with the control in the same groups.

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
