# Peer review of "Long-Term Exposure to Benzo[a]Pyrene Affects Sexual Differentiation and Embryos Toxicity in Three Generations of Marine Medaka (Oryzias Melastigma)"

_ijerph, 2020, doi:10.3390/ijerph17030970_

Round 1

Reviewer 1 Report

The study demonstrates the need to generate knowledge about the effects of BaP on the reproduction of various marine organisms such as medaka. However, the abstract and introduction should be improved since its wording is confusing. In addition, the authors are advised to avoid the claim that BaP is a hormone, as its effects resemble its responses, but a hormone works in a regulated manner and this compound has a nonspecific behavior. Finally, it is suggested that the discussion of the findings of this study be improved.

Author Response

Q: The study demonstrates the need to generate knowledge about the effects of BaP on the reproduction of various marine organisms such as medaka. However, the abstract and introduction should be improved since its wording is confusing. In addition, the authors are advised to avoid the claim that BaP is a hormone, as its effects resemble its responses, but a hormone works in a regulated manner and this compound has a nonspecific behavior. Finally, it is suggested that the discussion of the findings of this study be improved.

Answer: Thanks for your advice. We invited native-speaking English experts to polished the abstract, introduction, and discussion sections. And, we modified "BaP is a hormone" to "BaP is an endocrine-disrupting chemical ".

Reviewer 2 Report

Long-term exposure to Benzo[a]pyrene affects sexual differentiation and embryos toxicity in three generations of marine medaka (Oryzias melastigma)

Comments:

Benzo[a]pyrene (BaP) is a common environmental hormone that can cause endocrine disorders in organisms. In this study, they used three generations of marine medaka to test the influence of BaP to the embryo toxicity, and the sexual differentiation. They also found that BaP has significantly inhibited the expression of the genes related to female fish development. Thet concluded BaP can cause embryonic toxicity and abnormal sexual differentiation, while the expression of related reproductive genes can effectively indicate the sex ratio.

The manuscript is in general well written, but this reviewer recommends the authors to revise it. There are some comments below, which this reviewer recommends to be addressed by the authors.

Comments:

In Table 2. Many datas’ error bars are 0.58. Could you upload the original data? In Table 3. Could you explain why the sexual maturity time of G3F3 is shorter than G2F3 when cultured in 2ug/L BaP? What is your opinion on BaP influence? Will the fish get resistance to BaP after it cultures in medium with BaP for a period? What is the reason that G2F2 performs the same as G1F2 except when BaP concentration was increased to 200 ug/L (Table 3.)? What is the concentration of BaP in our environment? What do you think that G2F3 have longer the sexual maturity time than G2F2? Do you measure the BaP concentration in embryos? According to your Figure 3, it seems that the concentrations of BaP used in this study did not have consistent trend of influence to the same generation, such as ERa expression in G1F3, cyp19a in G2F3. Could you explain? Does people have other aspects to consider for the influence of BaP except those you mentioned in this study?

Author Response

Comments:

Q: In Table 2. Many datas’ error bars are 0.58. Could you upload the original data?

Answer: Upload the original data of “hatching time” excel file.

Q: In Table 3. Could you explain why the sexual maturity time of G3F3 is shorter than G2F3 when cultured in 2ug/L BaP?

Answer: This phenomenon may be due to long-term exposure to low concentrations of BaP, marine medaka appeared typical of hormesis, which has long been BaP stress, there may be have the effect of overcorrection or excessive compensation.

Q: What is your opinion on BaP influence?

Answer: BaP can interfere with the endocrine regulation of organism, not only the teratogenicity and carcinogenicity of organism. In this study, the marine medaka continuously exposed to BaP showed different conditions and degrees of response. Therefore, we believe that BaP has a significant endocrine disrupting effect, which may cause irreparable effects on organisms and even humans for a long time exposure.

Q: Will the fish get resistance to BaP after it cultures in medium with BaP for a period?

Answer: Yes, marine medaka were resistant to long-term BaP exposure. However, not all concentration groups can produce resistance. According to the results of sex ratio and growth (upload the growth original data), resistance was produced at 20 μg·L-1, but not at 2 and 200 μg·L-1. It may be that high concentration of BaP (200 μg·L-1) has destroyed the self-regulation mechanism of marine medaka, while 2 μg·L-1 of BaP has not significantly activated the self-regulation of marine medaka.

Q: What is the reason that G2F2 performs the same as G1F2 except when BaP concentration was increased to 200 ug/L (Table 3.)?

Answer: It may be because BaP at low / medium concentrations (2 and 20 μg·L-1) does not have too obvious generation characteristics for the sexual maturation time of marine medaka.

Q: What is the concentration of BaP in our environment?

Answer: The concentration of BaP is less than 1 μg·L-1 in most studies. However, some studies have shown that the BaP concentration in the environment can reach 96.8 μg·L-1[1], and even some polluted water bodies can reach the 1.239 mg·L-1[2].

Maskaoui K, Zhou J L, Hong H S, et al. Contamination by polycyclic aromatic hydrocarbons in the Jiulong River estuary and Western Xiamen Sea, China[J]. Environmental pollution, 2002, 118(1): 109-122. Edokpayi J N, Odiyo J O, Popoola O E, et al. Determination and distribution of polycyclic aromatic hydrocarbons in rivers, sediments and wastewater effluents in Vhembe District, South Africa[J]. International journal of environmental research and public health, 2016, 13(4): 387.

Q: What do you think that G2F3 have longer the sexual maturity time than G2F2?

Answer: Although G2F3 is an unexposed treatment group (non-BaP), the first two generations were exposed to BaP, and the enrichment effect occurred with the continuation of generations, which showed an enhanced trend in F3 generations. Therefore, the sexual maturity of G2F3 Longer than G2F2.

Q: Do you measure the BaP concentration in embryos?

Answer: We did not measure BaP in embryos. This study will be supplemented in subsequent experiments.

Q: According to your Figure 3, it seems that the concentrations of BaP used in this study did not have consistent trend of influence to the same generation, such as ERa expression in G1F3, cyp19a in G2F3. Could you explain?

Answer: From the overall data, BaP has a significant anti-estrogen or androgenic effect. We have also observed that the results you mentioned are inconsistent at 20 μg·L-1, so we speculate that marine medaka may have the ability to self-regulate at 20 μg·L-1. However, this self-regulating ability is not stable, and there is a possibility of overcompensation.

Q: Does people have other aspects to consider for the influence of BaP except those you mentioned in this study? 

Answer: Most research focuses on the significant teratogenicity and carcinogenicity of BaP and how to eliminate this negative effect. Other studies have also focused on the endocrine effects and osteogenic effects of BaP. However, there are few studies on the effects of BaP on multi-generation organisms, especially in terms of sexual differentiation. This study also used new marine model organisms, marine medaka, as research objects, which better revealed the stress of BaP on marine organisms and the physiological and ecological response of marine organisms to BaP exposure.

Round 2

Reviewer 2 Report

accept